# Enhancing Infrared Vision: Progressive Prompt Fusion Network and Benchmark

**Jinyuan Liu**[†], **Zihang Chen**[†], **Zhu Liu**[†], **Zhiying Jiang**[‡], **Long Ma**[†], **Xin Fan**[†], **Risheng Liu**[†*]

[†]School of Software Engineering, Dalian University of Technology
[‡]Information Science and Technology College, Dalian Martime University
atlantis918@hotmail.com, chenzi_hang@mail.dlut.edu.cn

## Abstract

We engage in the relatively underexplored task named thermal infrared image enhancement. Existing infrared image enhancement methods primarily focus on tackling individual degradations, such as noise, contrast, and blurring, making it difficult to handle coupled degradations. Meanwhile, all-in-one enhancement methods, commonly applied to RGB sensors, often demonstrate limited effectiveness due to the significant differences in imaging models. In sight of this, we first revisit the imaging mechanism and introduce a Progressive Prompt Fusion Network (PPFN). Specifically, the PPFN initially establishes prompt pairs based on the thermal imaging process. For each type of degradation, we fuse the corresponding prompt pairs to modulate the model's features, providing adaptive guidance that enables the model to better address specific degradations under single or multiple conditions. In addition, a Selective Progressive Training (SPT) mechanism is introduced to gradually refine the model's handling of composite cases to align the enhancement process, which not only allows the model to remove camera noise and retain key structural details, but also enhancing the overall contrast of the thermal image. Furthermore, we introduce the most high-quality, multi-scenarios infrared benchmark covering a wide range of scenarios. Extensive experiments substantiate that our approach not only delivers promising visual results under specific degradation but also significantly improves performance on complex degradation scenes, achieving a notable 8.76% improvement. Code is available at https://github.com/Zihang-Chen/HM-TIR.

## 1 Introduction

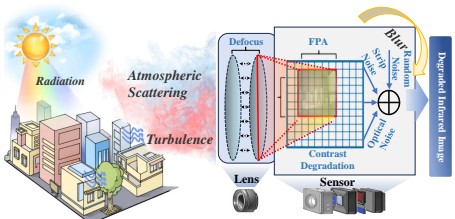

Figure 1: An illustration of the thermal infrared degradation pipeline. Thermal infrared imaging is prone to degradation from external factors such as solar radiation, atmospheric scattering, and turbulence, as well as internal factors like pixel size, internal noise, and jitter.

Thermal Infrared (TIR) imaging captures images by detecting the thermal radiation emitted by objects, typically within the wavelength range of 8 to 14 micrometers. Unlike visible light imaging, TIR does not depend on external light sources, allowing it to function effectively in complete darkness or low-light conditions. Its ability to penetrate smoke, haze, and minor obstructions, coupled with accurate temperature data, makes TIR essential for diverse applications [69, 67], such as object detection [32, 35], semantic segmentation [68], and autonomous driving [31].

Despite its advantages, TIR imaging faces significant challenges that limit its widespread use. The com-

---

[*]Corresponding Author

39th Conference on Neural Information Processing Systems (NeurIPS 2025).

plexity of the imaging process and the reliance on expensive, specialized materials like Mercury Cadmium Telluride (MCT) and Indium Antimonide (InSb) make obtaining high-quality TIR images difficult. Additionally, TIR systems are highly susceptible to external factors such as temperature fluctuations and varying atmospheric conditions, which can degrade image quality. These obstacles underscore the critical need to advance thermal infrared image enhancement techniques.

A considerable number of image enhancement methods have been proposed for TIR or visible images. Techniques such as histogram equalization [48], adaptive filtering [49, 38], and deep learning-based approaches [23, 5, 12, 36] have been utilized to improve image contrast, reduce noise, and enhance overall visual quality. However, these methods exhibit two major limitations. Firstly, enhancement techniques developed for visible images often prove challenging to apply to TIR images due to fundamental differences in imaging modalities, degradation and imaging processes. Secondly, existing enhancement methods only address single degradation, such as denoising or encontrast.

Moreover, a major obstacle in TIR image enhancement is the limited availability of diverse datasets. Although learning-based ways have demonstrated success in various image processing applications, they require large and varied datasets to train effectively and generalize well. However, existing datasets encompass only a narrow range of scenes and conditions, making it challenging to validation.

Incorporating these criteria, this paper presents the Progressive Prompt Fusion Network (PPFN) for enhancing TIR images. PPFN comprises two key components: type and degradation-specific prompts and a prompt fusion module. The degradation-specific prompts guide the model in identifying degradation types, while type-specific prompts differentiate single from composite degradation scenarios. The prompt fusion module integrates prompt pairs to iteratively modulate model features, providing adaptive guidance tailored to specific degradation types in both single and multiple contexts. Additionally, we introduce a Selective Progressive Training (SPT) mechanism for handling composite and single degradations, which iteratively refines each degradation step by using the output from one stage as input for the next in composite scenarios, while applying standard training for single degradations. Consequently, the model effectively eliminates each impairment without interference, resulting in significant performance improvements. Our contributions can be summarized into four key aspects, as follows:

- We propose a PPFN to enhance TIR images, delivering exceptional visual quality in hybrid degradations. To our knowledge, this is the first study addressing TIR enhancement under such multifaceted degradation conditions.

- Addressing intricate degradations in real-world thermal infrared images, we introduce a prompt fusion block that incorporates prior knowledge into the learning process, effectively managing both single and hybrid degradations. Importantly, the prompt fusion block is a plug-and-play module that seamlessly integrates into various existing network architectures, enhancing performance.

- We propose a SPT scheme that optimizes both single and hybrid degradation scenarios, enabling the model to effectively refine complex degradations while ensuring robustness and stability under simpler conditions.

- We establish a high-quality TIR benchmark covering multiple scenarios, named HM-TIR, with all collected images meticulously focused for clarity. This dataset encompasses diverse environments, including urban areas, forests, and oceans, to name a few.

## 2 Related Work

This section provides a concise overview of existing TIR and visible image enhancement techniques relevant to our study, as well as the necessary benchmarks for learning and empirical evaluation.

### 2.1 TIR/Visible Image Enhancement

With the growing demands of modern applications, numerous TIR image enhancement methods have been developed, achieving promising results. For TIR denoising, studies [33, 5, 2] have simulated realistic infrared noise by combining various noise types, resulting in significant improvements. Additionally, researchers have addressed specific blur types, including motion blur [58, 17], out-of-focus blur [71], and Gaussian blur that simulates atmospheric effects [62]. These efforts have

substantially enhanced image clarity and detail restoration in infrared imaging. TIR are also vulnerable to other degradations, such as compression artifacts and low resolution. Several studies [1, 16, 28, 29] have tackled these challenges, leading to notable advancements. However, existing methods are typically constrained by specific degradation conditions, which significantly limits their generalization and effectiveness in real-world infrared image processing.

Table 1: Illustration of our benchmark and existing infrared enhancement datasets. The "multiplication" denotes the diverse camera viewpoints, including horizontal, surveillance, driving, etc.

| Scene: ①: Road ②: Square ③: City ④: Forest ⑤: Campus ⑥: Coastline ⑦: Residential Zone ⑧: Others | | | | | | | |
|---|---|---|---|---|---|---|---|
| Corruption: I: Low Contrast | | II: Blur | III: Stripe Noise | IV: Optical Noise | | V: Gaussian Noise | |
| Dataset | Year | Format | # of Images/Videos | Resolution | Camera angle | Scene | Corruption Type |
| EN [23] | 2019 | Image | 16 | 256×256 | horizontal&surveillance | ④⑤⑦ | I |
| Iray [34] | 2021 | Image | 2000 | 256×192 | horizontal | ①⑧ | III |
| SBTI [25] | 2022 | Video | 4 | 640×480 | horizontal&surveillance | ①③ | II |
| UIRD [20] | 2023 | Video | 17 | 640×512 | horizontal&surveillance | ①③ | II |
| TIVID [2] | 2024 | Video | 518 | 320×256 | horizontal | ①③④⑦ | III IV V |
| HM-TIR (Ours) | 2025 | **Image** | **1503** | **640×512** | **multiplication** | ①∼⑧ | **I∼V** |

All-in-One Image Restoration employs a single model to address a range of image degradation issues. PromptIR [45] and ProRes [39] use additional degradation context to introduce task information. IDR [61] explores the model optimization by ingredient-oriented clustering. AutoDIR [18] leverages latent diffusion with degradation-specific text embeddings to automate degradation handling. InstructIR [9] introduces natural language instructions to control restoration. However, most of these methods are only focus visible image enhancement, posing a chanllenge to apply in TIR images.

## 2.2 Thermal Image Enhancement Benchmarks

In recent years, several image enhancement benchmarks addressing specific degradations have been introduced, including the Iray Infrared Image Denoising dataset [34] and TIVID [2] for thermal image denoising, EN [23] for contrast enhancement, and SBTI [25] and UIRD [20] for deblurring. The Iray dataset comprises 2,000 pairs of real-world noisy infrared images captured indoors and outdoors alongside their clean counterparts. TIVID includes 518 diverse videos collected with a cooled infrared imaging system to simulate various thermal infrared noises. EN contains 16 internet-sourced images designed to evaluate contrast enhancement. Additionally, SBTI consists of four videos captured on roads and around vehicles, while UIRD includes 17 videos generated through frame interpolation to produce more blurred images.

Table 1 outlines the main attributes of these datasets, including scale, resolution, lighting conditions, and scenario types. Limited resolution, quality, and scenario, degradation types, and overall dataset size variety restrict their applicability for real-world infrared enhancement tasks.

## 3 Methodology

### 3.1 Problem Formulation

Infrared imaging systems, especially those using CMOS-based sensors, are prone to additional Fixed-Pattern Noise (FPN) alongside random noise types common in RGB imaging, such as Gaussian and salt-and-pepper noise. Additionally, unlike visible images, which contain detailed visual content and high-quality representation, infrared images capture only thermal distributions, making them particularly vulnerable to atmospheric conditions and temperature differences. As illustrated in Figure 1, this degradation pipeline is affected by several factors, including low contrast due to minimal temperature differences, blurring from environmental radiation effects, and sensor-induced noise, which collectively reduce image clarity and quality. We categorize TIR degradation into three primary types: low contrast, blurring, and noise. The degradation process unfolds in a specific sequence: low contrast occurs first, followed by blurring, and ending with noise.

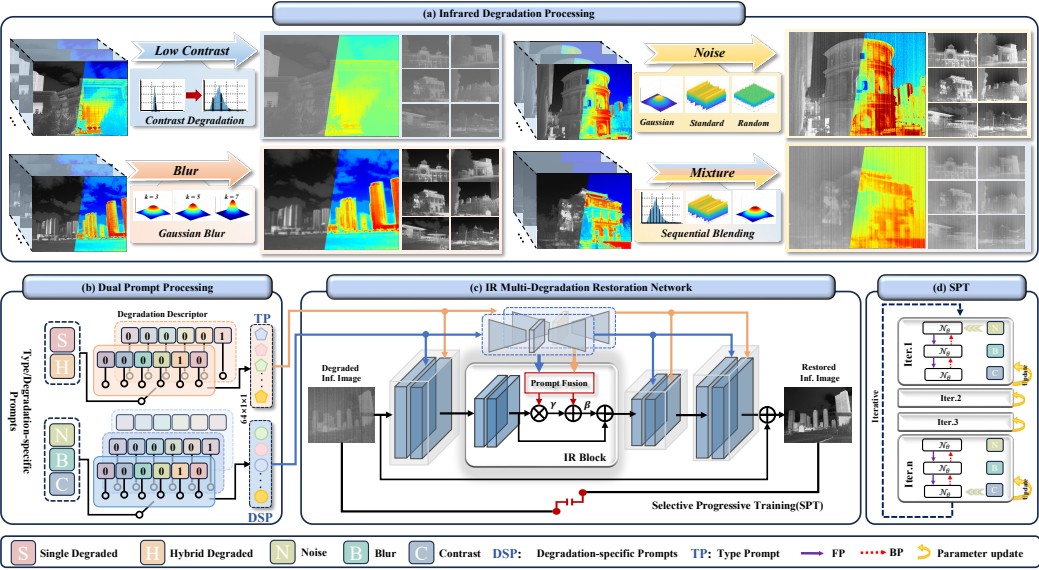

Figure 2: Schematic diagram of the proposed TIR enhancement framework. In subfigure (a), we first illustrate the TIR degradation process, including low contrast, blur, and noise across single and composite degradation scenarios. Subfigures (b) and (c) present details of the PPFN integrated with the image restoration model. Lastly, we depict our SPT in subfigure (d).

Therefore, given an observed clean image $\mathbf{I}_c$, the degraded image $\mathbf{I}_d$ can be formulated as:

$$\mathbf{I}_d = \left( \underbrace{\mathbf{n}_s \circ \mathbf{n}_o}_{\text{FPN}} \circ \underbrace{\mathcal{K}}_{\text{Blur}} \circ \underbrace{\mathcal{C}}_{\text{Low Contrast}} \right) (\mathbf{I}_c) + \mathbf{n}_r, \tag{1}$$

where $\mathcal{C}$, $\mathcal{K}$, $\mathbf{n}_o$, $\mathbf{n}_s$, and $\mathbf{n}_r$ represent the degradation with low contrast, blur kernel, optics, stripe, and additive random noise, respectively. $\circ$ denotes the composition operation.

As shown in Eq. (1), TIR degradation encompasses multiple types that strongly impact TIR images. To enable the base enhancement model to address various degradations in both complex composite and single degradation scenarios, we propose a prompt fusion learning strategy, as described in Sec. 3.2. Furthermore, to improve the model's stability in addressing composite degradations, we introduce the Selective Progressive Training strategy, as described in Sec. 3.3.

### 3.2 Prompt Fusion Learning

The primary challenge in infrared image enhancement is the diverse range of degradation types, which single models cannot effectively address. Existing networks typically target specific degradations and struggle with complex composite ones. Although all-in-one restoration frameworks aim to remove multiple degradation types, they often falter with intricate composite degradations. To overcome this, we introduce the Progressive Prompt Fusion Network (PPFN), which enhances image restoration models for more effective infrared enhancement in complex scenarios. As shown in Figure 2(b) and (c), our PPFN comprises a dual-prompt processing module and a prompt fusion module.

In dual prompt processing, we introduce type-specific and degradation-specific prompts. The degradation-specific prompt $\mathbf{P}_{deg} := \{\mathbf{p}_{deg}^n, \mathbf{p}_{deg}^b, \mathbf{p}_{deg}^c\}$ guides the model to adapt degradation types, while the type-specific prompt $\mathbf{P}_{type} := \{\mathbf{p}_{type}^s, \mathbf{p}_{type}^h\}$ is utilized to enable the model to distinguish the difference between single and composite degradation scenarios. Here, $n$, $b$, and $c$ represent noise, blurring, and contrast degradation, respectively, while $s$ and $h$ denote single and composite degradation scenarios. The degraded images processed by each step in either single or composite degradation scenarios with specific prompts $\mathbf{p}_{deg}^i \in \mathbf{P}_{deg}$, $i \in \{n, b, c\}$ and $\mathbf{p}_{type}^j \in \mathbf{P}_{type}$, $j \in \{s, h\}$. To extract prompt features, we first obtain the degradation-specific prompt feature $\mathbf{F}_{deg}^p$ and type-specific prompt feature $\mathbf{F}_{type}^p$ using two lightweight prompt encoders,

$\mathbf{E}_{deg}$ and $\mathbf{E}_{type}$, which are expressed as:

$$\begin{aligned}
\mathbf{F}_{deg}^p &= \mathbf{E}_{deg}(\mathbf{p}_{deg}^i), \ i \in \{n, b, c\}, \\
\mathbf{F}_{type}^p &= \mathbf{E}_{type}(\mathbf{p}_{type}^j), \ j \in \{s, h\}.
\end{aligned} \quad (2)$$

To represent the prompt more efficiently and guarantee subsequent injection being a conventional modulation manner [27, 22], a prompt fusion module is introduced. Specifically, we concatenate the two prompt features and then apply a linear layer $\mathcal{W}_{fusion}$, followed by a non-linear activation $\phi(\cdot)$, to obtain the final prompt feature $\mathbf{F}_p$, as expressed below:

$$\mathbf{F}_p = \phi(\mathcal{W}_{fusion}(\texttt{Cat}(\mathbf{F}_{deg}^p, \mathbf{F}_{type}^p))), \quad (3)$$

where the operator $\texttt{Cat}(\cdot, \cdot)$ denotes concatenate operation. To integrate the prompt into the model's feature space and enable adaptability across degradation and scenario type, we calculate two channel-wise modulation parameters with suitable dimension, $\boldsymbol{\gamma}$ and $\boldsymbol{\beta}$, by applying a linear layer $\mathcal{W}_p$,

$$\boldsymbol{\gamma}, \boldsymbol{\beta} = \mathcal{W}_p(\mathbf{F}_p). \quad (4)$$

Given the $l$-th layer feature $\mathbf{F}_l \in \mathbb{R}^{h_i \times w_i \times c_i}$ in restoration model, with calculated modulation parameters $\boldsymbol{\gamma}_l \in \mathbb{R}^{1 \times 1 \times c_i}$ and $\boldsymbol{\beta}_l \in \mathbb{R}^{1 \times 1 \times c_i}$, this adaptation process can be expressed as follows:

$$\tilde{\mathbf{F}}_l = \mathbf{F}_l \otimes (1 + \boldsymbol{\gamma}_l) + \boldsymbol{\beta}_l, \quad (5)$$

where $\tilde{\mathbf{F}}_l$ is the updated model feature that will be passed to the next model block. By integrating PPFN module, the model enables more effective handling of composite degradations.

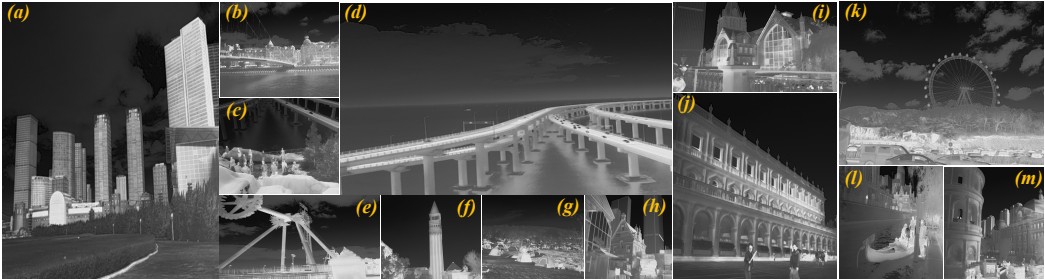

Figure 3: Example images from our HM-TIR benchmark include: (a) skyscraper, (b) seaside, (c) mountain, (d) cross-sea bridge, (e) pendulum, (f) tower, (g) camping area, (h) commercial street, (i) mansion, (j) square, (k) Ferris wheel, (l) boats, and (m) tourist attraction. More examples are provided in the *Supplementary Material A.3*.

### 3.3 Selective Progressive Training

To address the distinct challenges of composite and single degradations in TIR enhancement, we introduce the Selective Progressive Training (SPT) mechanism, as described in Figure 2(d). SPT refines the degradation process by progressively enhancing each iteration through feedback loops. For composite degradations, where steps are applied sequentially, each iteration's output feeds into the next, enabling the model to learn and adapt to complex, interdependent degradation patterns. In contrast, for single degradations, where one type of degradation is present, a standard training framework is employed. Given a degradation process with $N$ steps, we generate a sequence degraded images $\mathbf{I}_D := \{\mathbf{I}_d^1, \mathbf{I}_d^2, \cdots, \mathbf{I}_d^N\}$ by using clean image $\mathbf{I}_c$ with corresponding to degradation-specific prompts $\mathbf{P}_{deg} := \{\mathbf{p}_{deg}^1, \mathbf{p}_{deg}^2, \cdots, \mathbf{p}_{deg}^N\}$. As shown in Figure 2(a), for single degradation scenario, each degraded image is generated by using specific degradation to $\mathbf{I}_c$. While for composite scenario, the $k$-th degraded image $\mathbf{I}_d^k$ is generated by specific degradation to $k - 1$-th degraded image $\mathbf{I}_d^{k-1}$. When training network, we set the initial input $\mathbf{I}_{in}^N = \mathbf{I}_d^N$ because the $N$-th degraded image contains all degradations in composite scenarios. Then network removes each degradation step in reverse order, enhancing the degraded images accordingly. For the $k$-th iteration of degradation removal training, given the input image $\mathbf{I}_{in}^k$, the restored output image $\mathbf{I}_{rest}^k$ is produced by the restoration model $\mathcal{N}_\theta$. GT for this iteration of the restoration model is defined as follows:

$$\mathbf{I}_{gt}^k = \begin{cases} \mathbf{I}_c, & \text{for single scenario,} \\ \mathbf{I}_d^{k-1}, & \text{for composite scenario.} \end{cases} \quad (6)$$

This setup ensures that only the $i$-th specific degradation is removed for both single and composite scenarios. Then we calculate the model loss gradient $\nabla_{\boldsymbol{\theta}} \mathcal{L}(\mathbf{I}_{rest}^k, \mathbf{I}_{gt}^k)$ but do not update the network parameters. This approach prevents the model from focusing excessively on any single type of degradation while potentially neglecting others, and ensures that the training sequence does not interfere with single scenario training. For the next iteration's input, if we use $\mathbf{I}_d^{k-1}$ directly in the composite scenario, the model will be affected by the removal of residual degradation from the previous iteration, leading to a significant drop in performance. To prevent this, we set the input $\mathbf{I}_{in}^{k-1}$ for the enhancement model in the next iteration (if it exists) as:

$$
\mathbf{I}_{in}^{k-1} = \begin{cases} \mathbf{I}_d^{k-1}, & \text{for single scenario,} \\ \mathrm{sg}(\mathbf{I}_{rest}^k), & \text{for composite scenario,} \end{cases} \tag{7}
$$

where $\mathrm{sg}(\cdot)$ denotes stop gradient operation to reduce training cost. After all iterations are completed, we update the model parameters using the sum of gradients computed across all iterations. In our TIR Enhancement setting, we define a three-step degradation process: noise, blur, and contrast. In the training phase, the degradations are added sequentially for composition scenarios: noise, blurring, and contrast. In the inference phase, we reverse this order to progressively remove the degradation: denoising, deblurring, and decontrast. The procedure is given in Alg. 1.

### 3.4 High-quality Multi-scenarios TIR Benchmark

Considering that limited diverse data has hindered the development of TIR domain, we establish a high-quality multi-scenario TIR benchmark, HM-TIR. It includes 1,503 TIR images encompassing various object types across different scenarios, as detailed in the last row of Table 1.

Each TIR image has a standard resolution of 640×512 and a wavelength range of 8 to 14 micrometers. To enhance thermal imaging performance by minimizing blur and increasing contrast, we individually adjusted the focus for each scene and secured the settings with mechanical tools before capturing. As shown in Figure 3, the HM-TIR benchmark includes a diverse structured environments, such as skyscrapers and Ferris wheels; unstructured settings like

---

**Algorithm 1** Selective Progressive Training.

**Require:** Clean infrared images with $\{\mathbf{I}_c\}$, a restoration Network $\mathcal{N}_{\boldsymbol{\theta}}$ and other necessary hyper-parameters.
1: **while** not converged **do**
2:     Generate $\mathbf{I}_D$ and $\mathbf{P}_{deg}$ by randomly $\mathbf{p}_{type}$;
3:     $\mathbf{I}_{in}^N = \mathbf{I}_d^N$;
4:     **for** $k = N, \ldots, 1$ **do**
5:         $\mathbf{I}_{rest}^k = \mathcal{N}_{\boldsymbol{\theta}}(\mathbf{I}_{in}^k, \mathbf{p}_{deg}^k, \mathbf{p}_{type})$;
6:         Set GT image $\mathbf{I}_{gt}^k$ according to Eq. (6);
7:         Calculate gradient $\nabla_{\boldsymbol{\theta}} \mathcal{L}(\mathbf{I}_{rest}^k, \mathbf{I}_{gt}^k)$;
8:         Set next input $\mathbf{I}_{in}^{k-1}$ according to Eq. (7);
9:     **end for**
10:    Update parameter $\boldsymbol{\theta}$ by gradient descent;
11: **end while**
12: **return** $\boldsymbol{\theta}^*$.

---

forests; and challenging scenarios like densely populated areas and small targets. We also incorporated various viewing angles, including aerial, eye-level, and low-angle. Additional data collection process and sensor equipment are provided in *Supplementary Material A.3*.

## 4 Experimental Results

### 4.1 Training Details

**Training and testing data.** In our experiments, we trained the TIR enhancement model on our HM-TIR dataset, which contains 1,503 TIR images encompassing diverse object types across various scenarios. We divided the dataset into 80% for training and 20% for validation, ensuring a balanced evaluation of our model's performance. For multi-degradation TIR enhancement testing, we created two validation subsets to enable a more detailed assessment: the Normal Set and the Hard Set. The Normal Set comprises images with lower levels of degradation, whereas the Hard Set includes images with more severe degradation. For single-degradation TIR enhancement testing, we applied the same settings as the Hard Set to create three separate single-degradation test subsets. The detailed degradation strategies and settings of degradation levels are provided in the *Supplementary Material A.1*.

**Training settings.** We use Restormer [60] as the baseline model for TIR enhancement to evaluate our proposed module and strategy. All models are implemented in PyTorch on four 4090D GPUs with

default settings. For the baseline model, we follow the Gated Degradation pipeline [65] to synthesize degradation, with the probabilities of all gates set to 0.8. During training, we adopt the L1 loss [56] and employ the Adam optimizer with parameters $\beta_1 = 0.9$ and $\beta_2 = 0.999$. Each model is trained with a batch size of 4, using random cropping and flipping with a patch size of $256 \times 256$. The initial learning rate is set to $8 \times 10^{-5}$ and decays to $10^{-6}$ following a cosine annealing schedule. Each model is trained for a total of 300 epochs.

**Evaluation metrics.** In this work, the Peak Signal-to-Noise Ratio (PSNR) and Structural Similarity Index Measure (SSIM) [56] are employed to assess the quality of the enhancement results under reference-based conditions. The PSNR and SSIM assess the quality of results primarily from the spatial dimension, with larger values indicating better results. For reference-free conditions, three no-reference Image Quality Assessment (IQA) metrics to evaluate image quality: NIMA [50], MUSIQ [19], and NIQE [41]. For NIMA and MUSIQ, higher values indicate better quality, while for NIQE, lower values are preferred.

### 4.2 Results on Multi-degradation TIR

To evaluate some TIR enhancement models, including WFAF [42], LRSID [4], and TSIRIE [44], as well as visible all-in-one restoration models such as DA-CLIP [37] and DiffUIR [70], we use the Normal Set to compare their TIR enhancement performance with our approach. Quantitative and qualitative comparisons are shown in Figure 4.

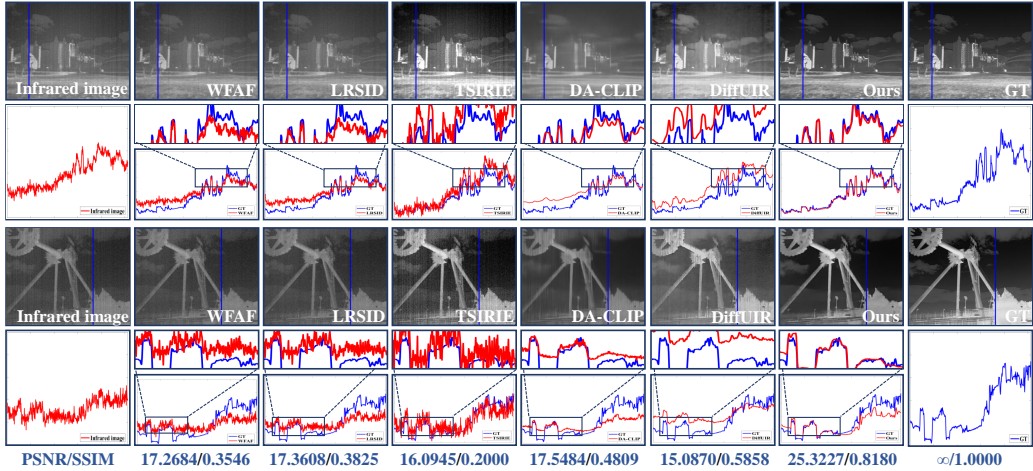

Figure 4: Quantitative and qualitative comparisons of signal performance across competitive image enhancement methods and our proposed approach. The average PSNR and SSIM values in our Normal Set are provided below the comparison figures in **blue**.

TIR enhancement methods such as WFAF, LRSID, and TSIRIE exhibit lower PSNR and SSIM values because they are tailored for single degradations and struggle with complex composite scenarios. In contrast, DA-CLIP and DiffUIR, developed as all-in-one enhancement methods for visible images, perform better; however, differences in imaging models between visible and infrared spectra lead to suboptimal results for infrared images. Our proposed method outperforms these approaches, achieving superior PSNR and SSIM scores and demonstrating enhanced signal restoration across multiple degradation scenarios. Qualitatively, traditional methods like WFAF, LRSID, and TSIRIE produce infrared images with substantial artifacts and background noise, while all-in-one approaches such as DA-CLIP and DiffUIR offer better restoration but still exhibit noticeable blurring and distortion. In contrast, our method excels at preserving critical structural information and fine details, reducing artifacts, and enhancing contrast.

We further evaluate our method alongside competitive approaches on the real-world Iray dataset [34] and adding an additional TIR enhancement method IE-CGAN [23]. Since it only provides the denoised result as ground truth, we use no-reference IQA metrics. Both quantitative and qualitative results are provided in Table 2 and Figure 5. Existing TIR enhancement methods struggle with complex scenarios, typically addressing only one type of degradation. Furthermore, all-in-one enhancement methods designed for visible images are ineffective in handling the specific degradations

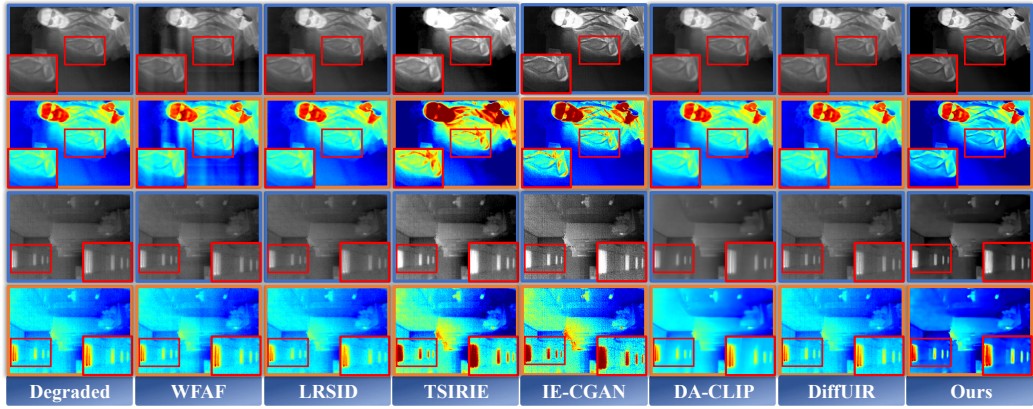

Figure 5: Qualitative comparisons of the competitive enhancement approaches and our method on Iray dataset.

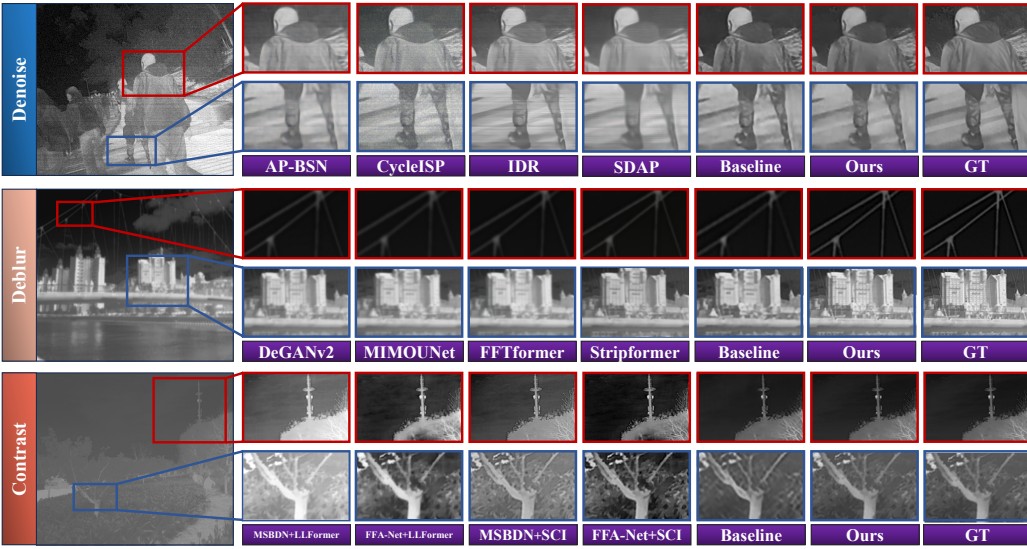

Figure 6: Qualitative comparison of single degradation between visible enhancement methods, baseline and our method on three single-degradation test sets, where "Baseline" refer to Restormer [60].

present in infrared image processing. In contrast, our approach not only outperforms existing methods in IQA scores but also shows superior restoration capabilities in real-world degradation conditions.

Table 2: Quantitative comparison in Iray dataset. The best is in red, and the second-best is in blue. "↓" means lower value is better.

| Metrics | Degraded | WFAF | LRSID | TSIRE | IE-CGAN | DA-CLIP | DiffUIR | Baseline | Ours |
|---------|----------|------|-------|-------|---------|---------|---------|----------|------|
| NIMA | 3.5326 | **3.7321** | 3.5682 | 3.5359 | 3.4959 | 3.7004 | 3.5935 | 3.5812 | **3.8327** |
| MUSIQ | 25.2459 | 25.1264 | 24.2095 | 23.7508 | **29.0350** | 27.7855 | 26.8066 | 27.7829 | **30.9072** |
| NIQE↓ | 10.1277 | 10.3536 | **8.6838** | 11.5204 | 9.4786 | 9.1896 | 9.3352 | 8.7776 | **8.4693** |

Due to page limitations, additional experimental results, *e.g.*, more comparisons in our test set and real-world datasets, are provided in *Supplementary Material A.4*.

## 4.3 Results on Single-degradation TIR

To evaluate three TIR enhancement models with single degradation scenarios, we conduct experiments in test subsets with denoising, deblurring, and contrast enhancement.

For denoising, we compare four state-of-the-art approaches: AP-BSN [26], CycleISP [59], IDR [66], and SDAP [43]. For deblurring, we evaluate leading methods including DeBlurGANv2 [24], MIMO-UNet [8], FFTformer [21], and Stripformer [52]. For contrast enhancement, treated as a combination of haze and low-light enhancement, we include MSBDN [13] and FFA-Net [46] for dehazing, alongside LLFormer [53] and SCI [40] for low-light enhancement. Using three single-degradation subsets, we compared the performance of these methods relative to ours, with qualitative comparisons shown in Figure 6. While existing methods effectively reduce degradation, they still retain artifacts due to modeling differences between TIR and visible images, resulting in lower enhancement quality and fidelity. In contrast, our method delivers superior performance in single-degradation TIR enhancement tasks, effectively reducing noise, recovering fine details, and enhancing contrast while preserving the natural appearance of images.

Table 3: Quantitative results comparing our method with five models in our two test sets, both with and without integration of our PPFN module and SPT strategy. 'Average' refers to the mean value across test sets. † denotes methods using our approaches to train. Baseline and our approaches results are shown in ■ and ■ boxes, respectively. The best result is in **red**, and the second-best in **blue**.

| Model | FocalNet PSNR/SSIM | FocalNet† PSNR/SSIM | UFormer PSNR/SSIM | UFormer† PSNR/SSIM | NAFNet PSNR/SSIM | NAFNet† PSNR/SSIM | XRestormer PSNR/SSIM | XRestormer† PSNR/SSIM | Restormer PSNR/SSIM | Restormer† PSNR/SSIM |
|---|---|---|---|---|---|---|---|---|---|---|
| Bridge | 22.45/0.832 | 24.95/0.865 | 22.75/0.850 | 24.10/0.844 | 24.88/0.859 | **25.98**/0.872 | **25.82**/0.879 | 25.14/**0.890** | 24.28/0.876 | 24.65/**0.896** |
| | 20.80/0.770 | **22.71**/0.803 | 20.93/0.780 | 20.95/0.778 | 19.87/0.758 | 22.44/0.786 | 21.52/0.775 | **23.52/0.812** | 21.12/0.783 | 22.27/**0.802** |
| Leaning Tower | 22.61/0.843 | 26.44/0.861 | 24.50/0.838 | 25.95/0.847 | 26.39/0.849 | 27.24/0.852 | 27.85/0.865 | **29.60/0.877** | 27.80/0.863 | **28.46/0.879** |
| | 22.89/0.801 | 21.21/0.806 | 18.82/0.723 | 17.72/0.724 | 23.25/0.778 | **23.49**/0.806 | 23.29/0.807 | **23.65/0.829** | 23.10/0.825 | 23.23/**0.832** |
| Tower | 24.89/0.858 | 24.33/0.851 | 23.20/0.838 | 25.71/0.863 | 25.78/0.864 | 26.60/0.867 | 26.46/0.874 | 27.86/0.893 | **29.29/0.903** | **30.40/0.908** |
| | 25.07/0.836 | 24.86/0.839 | 23.21/0.792 | 25.60/0.839 | 21.16/0.830 | 25.83/0.850 | 22.01/0.854 | **27.40/0.873** | 21.19/0.861 | 28.50/0.882 |
| Two Skyscrapers | 20.44/0.719 | 23.91/0.744 | 21.51/0.717 | 21.90/0.721 | 21.14/0.725 | 22.05/0.735 | 24.79/0.740 | 24.13/0.744 | **25.40/0.750** | 25.60/0.755 |
| | 23.10/0.687 | 24.38/0.700 | 21.96/0.665 | 22.81/0.675 | 23.16/0.671 | 22.57/0.693 | 24.57/0.696 | **26.05/0.709** | 24.71/0.701 | 26.08/0.716 |
| Villa | 22.28/0.779 | 26.00/0.855 | 25.17/0.809 | 23.66/0.817 | 24.10/0.815 | 27.70/0.851 | 27.86/0.854 | **29.06/0.865** | 22.46/0.785 | **29.73/0.871** |
| | 25.37/0.798 | 24.98/0.805 | 24.49/0.782 | 23.23/0.773 | 23.58/0.767 | 25.37/0.797 | **26.13**/0.798 | 25.71/**0.808** | 25.36/0.806 | 27.26/0.816 |
| Average | 21.22/0.778 | 22.63/0.790 | 21.95/0.775 | 21.62/0.768 | 22.29/0.776 | 23.74/0.792 | 23.54/0.801 | **24.75/0.811** | 23.28/0.796 | **25.32/0.818** |
| | 21.27/0.733 | 21.40/0.740 | 20.31/0.714 | 20.44/0.716 | 21.81/0.731 | 22.30/0.744 | 22.43/0.748 | **23.06/0.758** | 22.87/0.757 | **23.27/0.764** |

## 4.4 Ablation studies

**Validation of model architectures.** In addition to Restormer, we evaluate our PPFN module with four other SOTA image enhancement models: NAFNet [6], UFormer [55], XRestormer [7], and FocalNet [10]. We compare the performance of these five models with and without our PPFN module. Quantitative and qualitative results are presented in Table 3 and Figure 7, respectively.

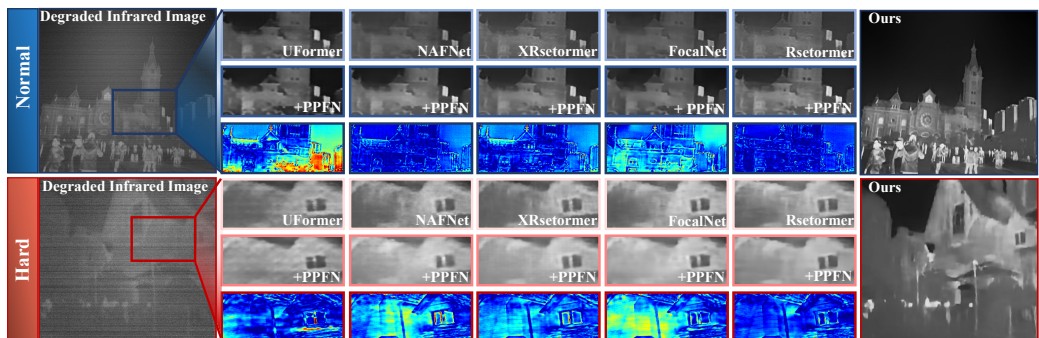

Figure 7: Visual comparison of five advanced methods with and without the integration of our approaches in Normal Set and Hard Set. Our method demonstrates superior visual quality and minimal error.

Quantitative and qualitative results show that all five baseline models exhibit lower PSNR and SSIM values and reduced enhancement quality on both Normal and Hard Sets, indicating their suboptimal performance with complex degradation. In contrast, integrating PPFN with each model consistently improves PSNR, SSIM, and TIR visual quality. Notably, our model achieves the best results, with an improvement of 8.76% on the Normal Set in PSNR.

**Study on prompt fusion learning.** We train models with the same settings as in previous comparison experiments. Testing is performed on the Hard Set, results are shown in Table 4. In dual prompt pro-

Table 4: Ablation studies on the PPFN and SPT strategy. The best is in **red**, and the second-best is in **blue**.

| # of Prompt | Prompt Fusion | Iter. | PSNR | SSIM |
|---|---|---|---|---|
| - | - | - | 22.8678 | 0.7568 |
| - | - | ✓ | 22.6357 | 0.7524 |
| DSP | - | ✓ | **23.1605** | 0.7635 |
| TP/DSP | w/o non-linear | ✓ | 23.1487 | **0.7646** |
| TP/DSP | Multiply | ✓ | 23.1432 | 0.7627 |
| TP/DSP | PPFN | 1 | 14.5455 | 0.6125 |
| TP/DSP | PPFN | 2 | 14.6080 | 0.6261 |
| TP/DSP | PPFN | 3 | **23.2712** | **0.7643** |

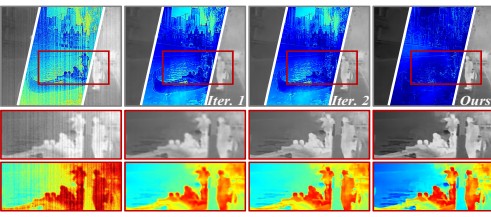

Figure 8: Analyzing the enhanced images and error maps from each iteration. Zoomed and pseudo-color maps for the best view.

cessing, applying degradation-specific and type/degradation-specific prompts achieves performance gains of 0.29 dB and 0.40 dB over the baseline, respectively. Regarding the prompt fusion strategy, removing non-linear activation or replacing the concatenation operation with multiplication results in a rapid PSNR decline, with the "Multiply" approach offering only minimal SSIM improvements. SPT reveals that directly applying iterative training to the baseline causes a PSNR drop of 0.23 dB.

**Analyzing the enhancement iteration.** We demonstrate the enhanced images from each iteration along with corresponding PSNR and SSIM values, as shown in Figure 8 and Table 4, respectively. Note that the iterations progress, specific degradations are incrementally removed, leading to a gradual improvement in both PSNR and SSIM metrics.

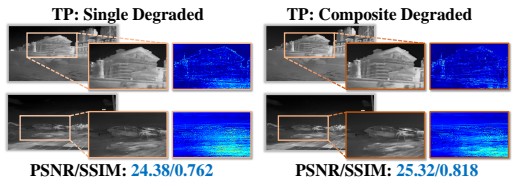

Figure 9: Quantitative and qualitative comparison of TIR enhancement performance between different type-specific prompt setting in Normal Set.

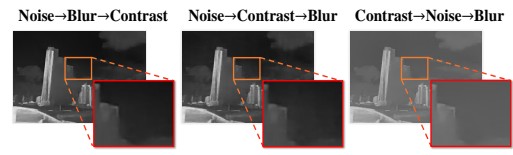

Figure 10: Quantitative and qualitative comparisons of our method with different order in each degradation removal process in Hard Set.

**Analyzing the prompt sensitivity.** We evaluate the performance of our method with incorrect prompts and order, as shown in Figure 9 and Figure 10. For incorrect prompts, we observe that using a single degradation scenario prompt with compositional degradation results in failure to remove degradation, with artifacts persisting. This indicates that the model struggles to eliminate residual degradation in each iteration under a single scenario. For incorrect order, we demonstrate that the model exhibits lower performance and PSNR when the degradation removal order is incorrect. This supports the hypothesis that optimal artifact removal occurs with a fixed processing order. Our results highlight that the SPT strategy effectively handles fixed-order degradation removal, leading to improved performance.

## 5 Conclusion

This paper introduced a new way for enhancing TIR images, managing complex degradation through dual-prompt processing and fusion modules. Our training scheme ensures robust performance across various scenarios. We also established a comprehensive TIR benchmark for accurate evaluation. Experiments show that PPFN surpasses existing methods in clarity, detail preservation, and contrast enhancement, advancing TIR image enhancement for broader applications.

## Acknowledgment

This work is partially supported by the National Natural Science Foundation of China (Nos.62302078, 62372080, 62450072, U22B2052, 624B2033), the Distinguished Youth Funds of the Liaoning Natural Science Foundation (No.2025JH6/101100001), the Distinguished Young Scholars Funds of Dalian (No.2024RJ002), the China Postdoctoral Science Foundation (No.2023M730741) and the Fundamental Research Funds for the Central Universities.

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

# A    Supplementary Material

In Supplementary Material, we present the degradation strategies and settings for degradation levels in Sec. A.1. Then we discuss the limitations of the current work in Sec. A.2. Also, we show additional details on our HM-TIR benchmark in Sec A.3. Additionally, we provide more experimental results in Sec. A.4. Finally, we provide a detailed discussions of our work in Sec. A.5.

## A.1    TIR Image Degradation Setting

In this subsection, we discuss the degradation simulation strategy and degradation level setting.

### A.1.1    TIR Degradation Simulation Strategy

**Low Contrast.** The raw pixel output of the detector quantizes the radiation response. These values are often unevenly distributed and confined to a narrow range due to small temperature differences [11], resulting in low-contrast images. Also, the suboptimal transformation methods may fail to map this limited range to a broader, visually distinguishable one, further reducing image interpret ability and utility. To simulate such degradation, we adopt a simple method that adjusts the unevenly and narrow distribution of the input TIR images $\mathbf{I}$, formulated as:

$$\mathcal{C}(\mathbf{I}) = \alpha \cdot (\mathbf{I} + \beta \cdot \mathrm{MAX}[\mathbf{I}]), \tag{8}$$

where $\alpha$, $\beta$, and $\mathrm{MAX}[\mathbf{I}]$ denote the reduction factor, offset factor, and the possible max pixel value of the image, respectively.

**Blur.** TIR images often suffer from blurring degradation due to various factors inherent to cameras and their surrounding environments. Two common types of blur in TIR images are low-pass blur and motion blur. Considering the availability of only single-frame TIR images, we focus exclusively on low-pass blur for simplicity.

Low-pass blur arises from atmospheric turbulence effects and the inherent limitations of camera capabilities, leading to loss of image details and reduced quality. To simulate this degradation, we follow prior works [72] and utilize an isotropic Gaussian blur and randomly select the kernel size and standard deviation of the kernel $k$ and blur the input TIR image, expressed as:

$$\mathcal{K}(\mathbf{I}) = k * \mathbf{I}, \tag{9}$$

where "$*$" denotes the convolution operation.

**Noise.** In TIR image processing, there are two types of noise mainly encountered: Fixed-Pattern Noise (FPN) and Random Noise.

**FPN** refers to the unique noise pattern characteristic of each digital camera. This phenomenon commonly arises when the camera is uncalibrated or affected by internal temperature fluctuations, and it is particularly pronounced in long-exposure shots. In TIR imaging systems, the most common types of FPN are Stripe Noise and Optics Noise.

**Stripe Noise** is a prevalent issue in TIR image processing, primarily resulting from amplification variations across the one-dimensional detector arrays in CMOS-based cameras [14]. Even after calibration, internal temperature fluctuations can exacerbate this phenomenon. This type of noise typically manifests as uneven horizontal or vertical stripe patterns. We assume the gain and offset of each detector unit to be approximately 1 and 0, respectively, and model them as two zero-mean Gaussian distributions with standard variances $\sigma_g$ and $\sigma_o$, respectively. To simulate degradation in TIR images, we randomly apply gain and offset along a single dimension, expressed as:

$$\mathbf{n}_s(\mathbf{I}) = (1 + g) \cdot \mathbf{I} + o, \tag{10}$$

where $g$ and $o$ represent gain and offset, sampled from two zero-mean Gaussian distributions, respectively.

**Optics Noise** arises from temperature response inconsistencies in TIR camera detector units. Prolonged use leads to non-uniform optics noise, influenced by internal temperature changes [3]. Two scenarios typically occur: during heating, the image center darkens while the edges brighten; during cooling, the center brightens while the edges darken. Following prior work [47], we model optics noise as a quartic cosine function of the distance between any pixel and the image center.

$$\mathbf{n}_o(\mathbf{I}) = \mathbf{I} + s_o \cdot \cos^4(\frac{\pi}{2} r(p, p_c)), \tag{11}$$

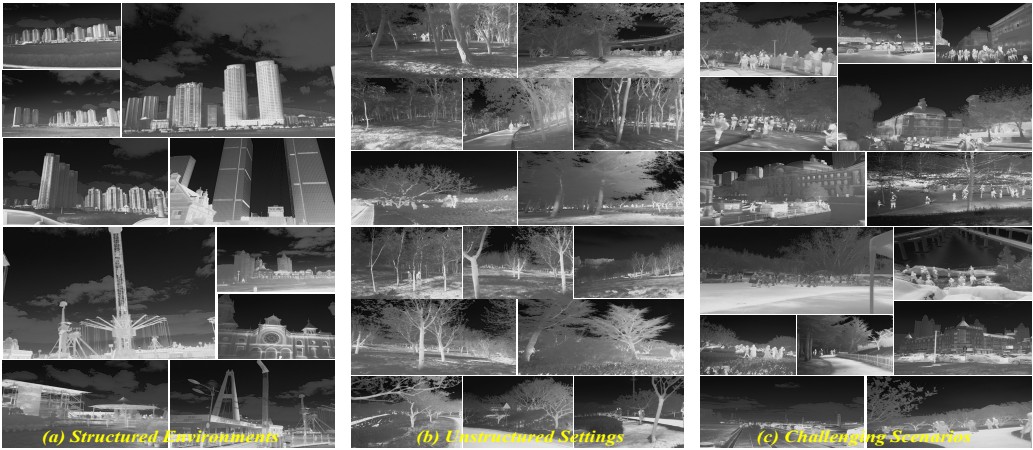

(a) Structured Environments  (b) Unstructured Settings  (c) Challenging Scenarios

Figure 11: Additional example images from our HM-TIR benchmark, including: (a) structured environment, (b) unstructured settings, (c) challenging scenarios.

where $s_o$, $p$, and $p_c$ represent the strength factor, current pixel position, and the center pixel position, respectively. $r(p, p_c)$ denotes the normalized distance, defined as:

$$r(p, p_c) = \frac{dist(p, p_c)}{\max(dist(p, p_c))},$$

(12)

where $dist(\cdot, \cdot)$ represents the distance function. We use the Euclidean distance to compute the separation between two points.

**Random Noise** is the result of several factors in TIR sensor sampling, *e.g.,* read noise and dark current in camera circuit [15]. It is generated by various factors and manifests as high-frequency random noise, which is not constant and can be described by statistical distributions. We utilize adding a white Gaussian noise $\mathbf{n}_r$ sampled from a Gaussian distribution with zero mean and standard variance $\sigma_r$ [30].

For noise addition order, we first consider optics noise $\mathbf{n}_o$, as it arises during the thermal radiation signal collection stage. Next, stripe noise $\mathbf{n}_s$ is introduced, as it occurs during the sensor production stage. Finally, additive Gaussian noise $\mathbf{n}_r$ is added at the last stage to represent random noise.

### A.1.2 TIR Degradation Level Settings

For degradation level settings, the simulator is controlled by eight parameters, enabling the generation of various types of degradation, including low contrast, blurring, and noise. For low contrast, the reduction factor $\alpha$ is set to range from 0.4 to 0.8, and the offset factor $\beta$ is set from 0.1 to 0.2 for the Normal Set. For the Hard Set, $\alpha$ ranges from 0.2 to 0.8, and $\beta$ ranges from 0.2 to 0.4. For blurring, the kernel size is varied from 7 to 17 and the standard deviation from 1 to 2 for the Normal Set, whereas for the Hard Set, the kernel size is adjusted from 7 to 23 and the standard deviation from 1 to 3. For noise, the standard deviation of gain $\sigma_g$ ranges from 0.03 to 0.07, the offset $\sigma_o$ varies from 0 to 3, the strength factor $s_o$ ranges from 15 to 55, and the standard deviation of white noise $\sigma_r$ ranges from 5 to 15 for the Normal Set. For the Hard Set, $\sigma_g$ varies from 0.03 to 0.10, $\sigma_o$ ranges from 0 to 5, $s_o$ spans from 15 to 75, and the standard deviation of white noise $\sigma_r$ ranges from 5 to 20.

### A.2 Limitations

Due to the inherent challenges associated with capturing paired degraded and clean TIR images, the degradation model employed in this study may not fully replicate the complexities of real-world degradation processes. The TIR imaging processing is typical susceptible to complex composite distortions, including motion artifacts, radiation attenuation, diffraction effects, and sensor-induced noise [64]. However, as demonstrated in Figure 5, 14, and Table 2, the proposed method achieves commendable performance in real-world TIR enhancement evaluations. These results provide strong validation for our degradation modeling strategy and support the effectiveness of the model under realistic conditions.

To overcome the limitation of the current approach, future work will aim to develop a more comprehensive degradation model that incorporates a broader range of noise and distortion types. This will improve generalization of TIR enhancement model, enabling more accurate real-world applications.

Table 5: Complexity comparison on parameters, FLOPs, and time.

| Methods | TSIRIE | DA-CLIP | DiffUIR | NAFNet | XRestformer | Baseline | Ours |
|---|---|---|---|---|---|---|---|
| *Params*(M) | 2.52 | 233.14 | 12.41 | 17.06 | 25.98 | 26.09 | 26.60 |
| Flops(G) | 77.91 | 660.18 | 164.68 | 79.47 | 820.52 | 704.10 | 704.33 |
| Time(s) | 0.01 | 17.07 | 0.325 | 0.024 | 0.348 | 0.292 | 0.876 |

### A.3    Additional Details of Benchmark

We have presented some examples of our HM-TIR benchmark in the main paper. In this section, we show additional examples of our benchmark in Figure 11. These HM-TIR benchmark example images include diverse conditions, such as structured environments, unstructured settings, and challenging scenarios. This highlights the high quality of our benchmark, which offers multi-scenario coverage and incorporates a diverse range of real-world challenges.

For the designed TIR prototype, we developed a TIR calibration board/algorithms to eliminate systematic errors/noise, and then reinforced the assembly to reduce environmental vibrations and added protective shields against electromagnetic interference. Besides, focus was adjusted for each scene, and post-processing included strict quality checks. In the post-processing stage, each image underwent strict quality checks to ensure reliability and high-quality. For non-cooled passive TIR sensor, we customized is a wavelength range of $8-14\mu m$, an aperture of $f/1.2$, HV-FOV of $48°\times38°$, Res.$640\times512$, and manual focus adjustment capabilities.

### A.4    More Experiments

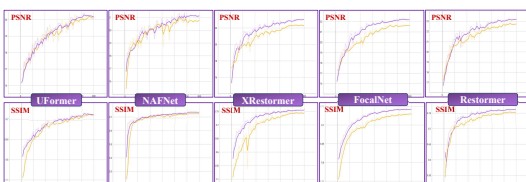

Figure 12: Comparison of PSNR and SSIM value curves during training across five models, with and without our approaches.

We provided some experiment results of our method on our test set and the Iray dataset in the main paper. In this section, we show the additional experiments. Firstly, we demonstrate more visual comparisons on the Normal Set and real-world Iray dataset, as shown in Figure 13 and Figure 14. These results further show that our approach has superior enhancement capabilities in our simulation scenarios and real-world degradation scenarios.

Then, we demonstrate that the PSNR and SSIM curves during training, shown in Figure 12, indicate that all five models achieve higher performance upon training completion with our PPFN module. This result demonstrates that our module and strategy adapt effectively, consistently enhancing the visual performance of each model.

In addition, we show the visual comparison of three baselines and their results with our strategies in TNO [51] and Roadscene [57], two real-world degraded TIR datasets, as shown in Figure 15 and Figure 16, respectively. It can be seen that the three baselines demonstrate limited enhancement performance, primarily reducing noise. In contrast, with our PPFN, models can generate more detailed outputs, reduce noise, and effectively enhance contrast.

Finally, we conduct complexity comparisons, as shown in Table 5. For fair evaluation, All the the models are equipped in hardware environment with a NVIDIA RTX 4090 D GPU with 24GB memory and the input TIR image resolution is 640×512. Our method, while introducing additional parameters and inference time due to prompt processing and stepwise degradation removal, outperforms baseline approaches in handling such complexities.

### A.5 Discussions

#### A.5.1 TIR Degradation Simulation Pipeline

The proposed TIR degradation simulation follows a fixed order. In contrast, some high-order degradation models have been extensively explored in the RGB domain, such as [54] and [63], realistic blur, noise, and compression artifacts are typically simulated by randomly or repeatedly applying multiple degradation operations, mimicking the effects of camera imaging, image editing, and Internet transmission. However, these simulation pipelines are inherently tailored to natural RGB images and fail to capture modality-specific degradations in TIR imaging, which are predominantly associated with the camera sensing process. Unlike RGB imaging, which relies on reflected light and is sensitive to illumination and weather, TIR imaging captures emitted thermal radiation and remains stable under varying conditions. However, due to its longer wavelength and sensor characteristics, TIR images suffer from unique degradations such as stripe noise, optical noise, and radiation-caused low contrast, especially in uncooled CMOS-based systems. These structured and composited degradations are uncommon in RGB images and cannot be effectively handled by RGB-oriented models.

#### A.5.2 Difference Between Cascade Multiple Specific Networks

Traditional methods, such as Cascade Multiple Specific Networks, address composited degradation by employing multiple independent sub-networks. In contrast, although our method performs iterative processing, it does not cascade multiple independent networks. Instead, it employs a unified network across iterations, modulated by degradation and scenario prompts, enabling progressive removal of each degradation type in different scenarios. This design avoids structural redundancy, and all iterations reuse the same network parameters, with only lightweight prompt modules introduced. While iterative inference may require more steps than single-pass baselines, the added cost is slight and results in significantly improved performance.

#### A.5.3 Prompt Design Detail

In our framework, two types of prompts are used as conditional inputs to guide the network. Following [22], these prompts are randomly initialized and fixed for each prompt type. During training, they gradually encode task-relevant conditions through model optimization under specific prompts. We manually define the prompt corresponding specific degradation conditions. Specifically, the prompts are designed to represent the degradation type and processing scenario. As part of future work, we plan to explore more flexible and scalable prompt designs. In particular, we will investigate the use of image-based self-prompting mechanisms, where the model dynamically generates degradation-aware prompts from the input itself. Additionally, we are interested in integrating language-based prompts to express complex degradation descriptions in a more interpretable and user-controllable manner.

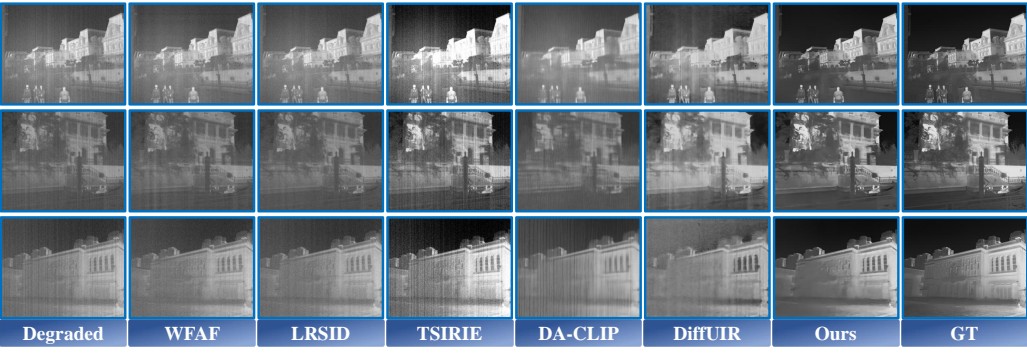

Figure 13: Additional visual comparisons of our method with other competitive approaches on our Normal Set.

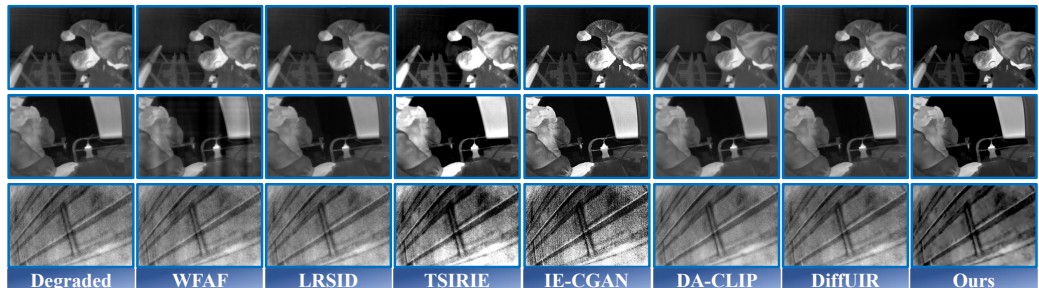

Figure 14: Additional visual comparisons of our method with other competitive approaches on Iray dataset.

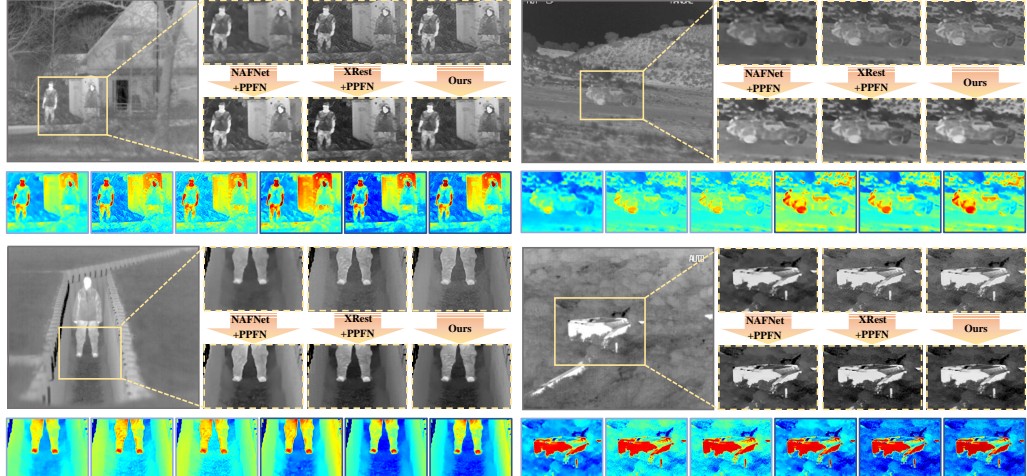

Figure 15: Visual comparisons of three baselines and with our PPFN approach in TNO dataset.

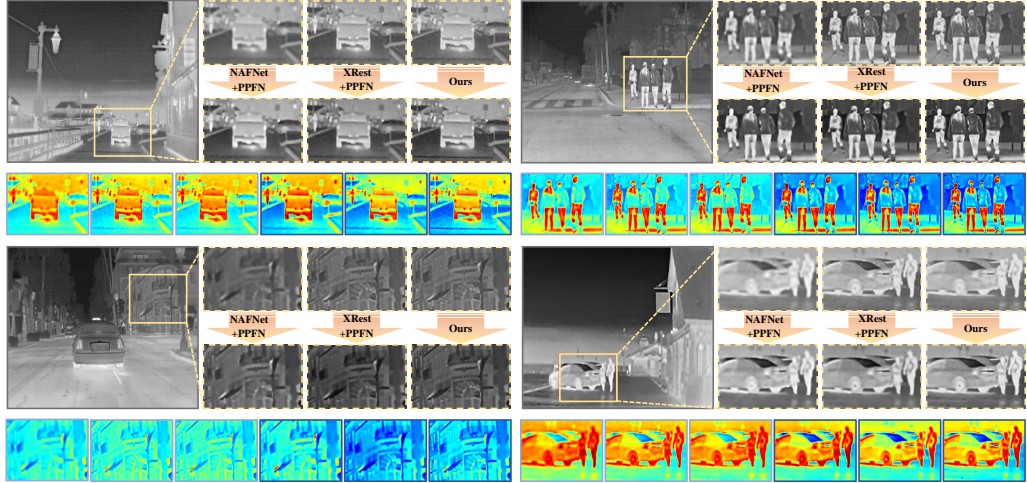

Figure 16: Visual comparisons of three baselines and with our PPFN approach in Roadscene dataset.

