# OpenReview forum: "Enhancing Infrared Vision: Progressive Prompt Fusion Network and Benchmark"
_NeurIPS.cc/2025/Conference — NeurIPS 2025 poster_

### Official Review · Reviewer_anvK · 2025-06-26

**Clarity:** 3
**Significance:** 3
**Originality:** 4
**Rating:** 6
**Confidence:** 5

**Summary:**

This paper introduces a novel Progressive Prompt Fusion Network (PPFN) alongside a Selective Progressive Training (SPT) strategy, aiming to enhance TIR images under complex composite degradation scenarios. Additionally, the authors construct a high-quality TIR dataset to facilitate model training and evaluation. Extensive experiments and ablation studies demonstrate that the proposed method outperforms SOTA approaches in both quantitative and qualitative metrics, while exhibiting superior robustness and generalization capability.

**Questions:**

Please refer to the weakness.

**Ethical Concerns:**

["NO or VERY MINOR ethics concerns only"]

**Final Justification:**

Thanks to the authors for their responses. The authors' rebuttal addressed my questions with a lot of efforts. In a such multifaceted complex degradation simulation, training models with TIR images and then working well in real-world is a huge advancement. Overall, this work is quite good and has laid the foundation for the following work.

**Limitations:**

Yes.

**Paper Formatting Concerns:**

There are no major formatting issues.

**Quality:**

4

**Strengths And Weaknesses:**

Strengths:
1. The figures are well-designed and effectively illustrate the core contributions of the paper, enhancing the clarity and readability of the proposed methods.
2. The proposed PPFN module and SPT strategy demonstrate strong capability in addressing complex degradation scenarios in infrared imagery, showcasing both innovation and practical effectiveness.
3. The introduction of the HM-TIR dataset is a valuable contribution, offering high-quality TIR images that span a diverse range of environmental conditions and real-world scenarios, which can benefit future research in thermal imaging area.

Weaknesses:
1. While the proposed TIR imaging degradation simulation is novel within the context of thermal imaging, high-order degradation models have been extensively explored in the general super-resolution literatures [1-2]. The authors are encouraged to explicitly clarify how their proposed TIR-specific degradation differences in existing high-order degradation in the RGB domain.
2. The related work section omits several recent advances in unified or all-in-one image restoration frameworks that address multiple degradation types within a single model. Incorporating a discussion of these works [3-5] would strengthen the contextual positioning of the proposed approach.
3. The paper contains some unclear representations that hinder understanding. Specifically, the meaning of "multiplication" in Table 1 is ambiguous, and the small font size and pics in Figure 6 affects readability and the clarity of key results.
4. The cost comparison analysis lacks critical implementation details, such as the hardware specifications (e.g., GPU model, memory capacity) and the resolution or size of the input images used during inference. These details are necessary to ensure a fair and reproducible comparison across different methods.

[1] Wang, Xintao, et al. "Real-esrgan: Training real-world blind super-resolution with pure synthetic data." ICCV Workshop. 2021.
[2] Zhang, Kai, et al. "Designing a practical degradation model for deep blind image super-resolution." ICCV. 2021.
[3] Potlapalli, Vaishnav, et al. "Promptir: Prompting for all-in-one image restoration." NeurIPS. 2023.
[4] Jiang, Yitong, et al. "Autodir: Automatic all-in-one image restoration with latent diffusion." ECCV. 2024.
[5] Conde, Marcos V, et al. "Instructir: High-quality image restoration following human instructions." ECCV. 2024.

---

> ### Author Rebuttal · Authors · 2025-07-31
>
> ## **Rebuttal to Reviewer anvK**
>
> We sincerely thank the reviewers for recognizing the novelty and significance of our work. Below, we provide clarifications and responses to address the main concerns.
>
> ### 1. **Clarification on the Difference Degradations Between TIR and RGB (W1):**
> We thank the reviewer for raising this important point. While high-order degradation models have been extensively studied in the RGB domain, thermal infrared (TIR) imaging exhibits fundamentally different physical characteristics and noise properties due to its reliance on thermal radiation rather than visible light. For RGB-based degradation models such as [1] and [2], realistic blur, noise, and compression artifacts are simulated by applying multiple degradation operations in a specific or random order, reflecting the effects of processes such as camera imaging, image editing, and Internet transmission. However, **these models are inherently designed for the general image and fail to capture the modality-specific degradations present in TIR images**, which mainly include the camera imaging. Specifically, **unlike RGB imaging, which relies on reflected light and is sensitive to illumination and weather, TIR imaging captures emitted thermal radiation and remains stable under varying conditions.** However, due to its longer wavelength and sensor characteristics, **TIR images suffer from unique degradations such as stripe noise, optical noise, and radiation-caused low contrast, especially in uncooled CMOS-based systems.** These structured and composite degradations are uncommon in RGB images and cannot be effectively handled by RGB-oriented models, **which typically focus on random noise, bad weather,  or low illumination and often fail under TIR scenarios, as shown in Figure 4 and 5.** We will revise the manuscript to incorporate these RGB high-order degradation works.
>
> ### 2. **Advanced Recent All-in-one Frameworks (W2):**
>
> Recent unified frameworks such as PromptIR [3], AutoDIR [4], and InstructIR [5] adopt various strategies for general image restoration: PromptIR uses implicit prompt tokens to improve the network in multi-task setting; AutoDIR leverages latent diffusion with degradation-specific text embeddings to automate degradation handling; InstructIR introduces natural language instructions to control restoration. While effective in the RGB domain, these methods still mainly focus on multiple single degradations, which are not designed for the composite, domain-specific degradations found in TIR images. **While our method employs a prompt-guided framework with a unified backbone and fusion of degradation and scenario-specific prompts.** This design enables explicit and progressively handling of diverse degradations across different scenarios. These all-in-one models will be incorporated into the revised version of the manuscript.
>
> ### 3. **Clarification of Ambiguous Representations and Figures (W3):**
>
> We acknowledge that certain notations and figures may have caused confusion. Specifically, the “multiplication” format is used to denote **diverse camera viewpoints, including variations across horizontal angles, surveillance views, and driving perspectives.** Additionally, we will increase the font size and enhanced the resolution of Figure 6 to improve readability.
>
> ### 4. **Supplementing Cost Comparison with Critical Implementation Details (W4):**
>
> To ensure fair and reproducible comparisons, we have supplemented the cost analysis section with detailed implementation information, including the hardware environment (NVIDIA RTX 4090 D GPU with 24GB memory) and the input TIR image resolution (640×512) used during inference.

---

> > ### Comment · Reviewer_anvK · 2025-08-05
> > **Comments after rebuttal**
> >
> > Thanks to the authors for their responses. The authors' rebuttal addressed my questions with a lot of efforts. In a such multifaceted complex degradation simulation, training models with TIR images and then working well in real-world is a huge advancement. Overall, this work is quite good and has laid the foundation for the following work.

---

### Official Review · Reviewer_VhxR · 2025-06-30

**Clarity:** 4
**Significance:** 4
**Originality:** 4
**Rating:** 5
**Confidence:** 4

**Summary:**

This paper aims to address the challenges associated with various and hybrid degradations in infrared imaging. It proposes a novel approach to model the degradation sequence, which includes contrast, blurring, and noise. The key innovation is the introduction of the Progressive Prompt Fusion Network (PPFN), a module designed to effectively mitigate specific degradations, whether they occur individually or in combination. Additionally, the Selective Progressive Training (SPT) mechanism is introduced, enabling the model to better handle hybrid degradation conditions. Extensive experiments demonstrate that the proposed PPFN outperforms state-of-the-art methods, showcasing its superior effectiveness in tackling infrared image degradation.

**Questions:**

Please refer to the weaknesses. In addition, I would like to ask the following questions for further clarification. 1. The degradation sequence modeling follows a fixed pipeline. Does the dynamic degradation modeling be more adaptive?  2. The meaning of the "multiplication" in camera angles in Table 1 is unclear. Could the authors clarify whether this refers to a combinatorial setup of camera angles or some other formulation?

**Ethical Concerns:**

["NO or VERY MINOR ethics concerns only"]

**Final Justification:**

The authors have already responded to my concerns.

**Limitations:**

No.  While the paper aims to address a wide range of degradation types in TIR domain, it appears that some extreme and practically relevant weather conditions, such as heavy rain, are not explicitly considered in the current degradation pipeline. It is recommended that the authors discuss this type of degradation in the paper, including whether it can be addressed by their current framework or if additional modeling would be required.

**Quality:**

4

**Strengths And Weaknesses:**

Strengths:
1. The proposed plug-and-play PPFN module and the hybrid degradation training strategy are novel contributions to the field of infrared imaging.
2. The newly introduced dataset is valuable and likely to benefit the broader infrared imaging research community.
3. The paper is clearly written and well-organized, with informative and well-designed figures that effectively support the presentation of the proposed methods and results.

Weaknesses:
1. The authors claim that “this is the first study addressing TIR enhancement under such multifaceted degradation conditions” However, to the best of my knowledge, several existing works have explored infrared enhancement under hybrid degradation settings [1-3]. Please make a detailed discussion with these works from degradation pipeline setting.
2. There is a lack of clarity in Table 1 regarding the dataset type used for each method. Specifically, whether the format is the single image or video frame. Please consider adding an additional row that explicitly indicates the dataset type for each method.
3. The text size in Figure 6 and Table 3 is too small, making it difficult to read, especially when printed or viewed on standard screens. Please revise these visual elements to improve readability and accessibility.

[1] Liu, Li, Luping Xu, and Houzhang Fang. "Simultaneous intensity bias estimation and stripe noise removal in infrared images using the global and local sparsity constraints." IEEE Transactions on Geoscience and Remote Sensing 58.3 (2019): 1777-1789.
[2] Zhang, Duo and Liu, Yinnian and Zhao, Yun, et al. "Algorithm research on detail and contrast enhancement of high dynamic infrared images." Applied Sciences 13.23 (2023): 12649.
[3] Cai, Lijing and Dong, Xiangyu and Zhou, Kailai, et al. "Exploring video denoising in thermal infrared imaging: physics-inspired noise generator, dataset and model." IEEE Transactions on Image Processing (2024).

---

> ### Author Rebuttal · Authors · 2025-07-31
>
> ## **Rebuttal to Reviewer VhxR**
>
> We sincerely thank the reviewer for their thorough review and constructive feedback. Below we address each of the raised concerns in detail.
>
> ### 1. **Clarification on the hybrid degradation in TIR enhancement (W1)**
>
> We thank the reviewer for this important observation. We agree that several prior works (e.g., [1-3]) have explored thermal infrared (TIR) image enhancement under certain hybrid degradation settings. For exmaple, [1] address stripe and optical noise, [2] focus on contrast and detail enhancement, and [3] target video denoising with a physics-inspired composite noise generator. **These methods typically operate under specific degradation setting, lack comprehensive degradation modeling,** and are not designed to generalize across diverse and dynamically composite degradation domains. While our work is the first to **systematically simulate, benchmark, and address comprehensive, real-world composite degradations in TIR images.**
> We will revise the manuscript to clarify our novelty and include a more detailed comparison with [1-3] in the Related Work section.
>
> ### 2. **Dataset type not clarified (W2)**
>
> We thank the reviewer for this valuable suggestion. To improve clarity, we will revise Table 1 by adding an additional row labeled “Input Type,” indicating whether each dataset is based on single images or video frames. For reference, **EN, Iray, and our proposed dataset are based on single images, while SBTI, UIRD, and TIVID are constructed from video sequences.**
>
> ### 3. **Readability issues (W3)**
>
> We appreciate the reviewer pointing this out. We will increase the font size of figure and table entries in the revised version to ensure that visualizations maintain clarity when printed or viewed digitally.
>
> ### 4. **Fixed Degradation Order Assumption (Q1)**
>
> **We argue that in TIR imaging, degradations typically follow a consistent order driven by imaging processes.** Low contrast comes first from limited thermal radiation, followed by blur from defocus or turbulence, and finally hybrid noise such as fixed pattern noise and random noise. Our framework follows this order, and we evaluate on real-world TIR images, **as shown in Figure 5 and Table 2, this design leads to strong performance in real-world scenes.** To test robustness, we also evaluated the model under incorrect degradation removal orders. As shown in Figure 10, performance drops significantly in some case, confirming the model works with a fixed processing order. In future work, we plan to model and predict degradation order to handle more complex scenarios.
>
> ### 5. **Ambiguity in “multiplication” notation (Q2)**
>
> Thank you for pointing out this ambiguity. The “multiplication” format is used to denote **diverse camera viewpoints, including variations across horizontal angles, surveillance views, and driving perspectives.** We will revise the table and its caption to clarify this interpretation explicitly.
>
> ### 6. **Coverage of extreme weather conditions (Limitation)**
>
> In Limitations section, we admit that our current degradation model does not fully cover all possible TIR degradations. Although we did not consider the extreme weather conditions, **we believe our framework retains some degree of generalization to such scenarios.** Specifically, in rainy or foggy environments, the increased scattering and absorption of thermal radiation reduce the energy reaching the sensor, especially from low-emissivity or distant objects. **This results in compressed temperature distributions and weaker radiation intensity, manifesting as low contrast.** Since our framework is trained to handle such contrast-degraded scenarios, it can partially generalize to these extreme weather-induced degradations, even without direct exposure during training. **Consequently, although our current model does not explicitly simulate all atmospheric effects like rain or fog.** We will clarify this point in the revised manuscript to highlight the potential of our approach to generalize beyond the explicitly simulated degradations.

---

> ### Comment · Reviewer_VhxR · 2025-08-04
>
> The authors have already responded to my concerns. I will maintain my score.

---

### Official Review · Reviewer_RuCv · 2025-06-30

**Clarity:** 4
**Significance:** 3
**Originality:** 4
**Rating:** 5
**Confidence:** 4

**Summary:**

This paper proposes a Progressive Prompt Fusion Network (PPFN) for enhancing infrared images with complex degradation types. Additionally, a high-quality, multi-scene benchmark dataset is constructed. Within PPFN, degradation-specific and type-specific dual prompts are introduced to identify degradation types and decouple composite degradations into individual ones. Subsequently, a plug-and-play prompt fusion block is incorporated to integrate prompts into the learning process. Finally, a Selective Progressive Training (SPT) mechanism is designed to iteratively handle different degradations. Extensive experiments demonstrate that the proposed method exhibits significant advantages in complex degradation scenarios.

**Questions:**

(1) Compared to “all-in-one” methods designed for visible images, what advantages does the proposed framework offer? Theoretically, this framework does not appear specifically tailored for infrared images.

(2) How does the iterative degradation removal framework fundamentally differ from cascading multiple degradation-specific networks? What are its distinctive advantages?

(3) During prompt processing, how are degradation-specific prompts and type-specific prompts acquired? Are they generated using other language models or manually annotated?

(4) Experiments on other datasets only provide visual results and lack quantitative analysis, which is insufficient to demonstrate the generalization capability of the proposed method.

**Ethical Concerns:**

["NO or VERY MINOR ethics concerns only"]

**Final Justification:**

I appreciate the author’s rebuttal, which has satisfactorily resolved my concerns. Accordingly, I have decided to increase my score.

**Limitations:**

Yes

**Paper Formatting Concerns:**

There are no major formatting issues.

**Quality:**

4

**Strengths And Weaknesses:**

Strengths:

(1) A high-quality infrared benchmark dataset covering diverse scenes is established, providing substantial support for advancement in this field.

(2) The proposed PPFN incorporates prompt learning and introduces an SPT mechanism, enabling iterative enhancement of infrared images with complex composite degradations.

(3) Extensive comparative and ablation experiments validate the effectiveness of the proposed method.

(4) The well-crafted figures and well-organized writing make the proposed approach easily understandable.

Weaknesses:

(1) In the Introduction, the authors state: “Firstly, enhancement techniques developed for visible images... imaging processes”. However, the proposed method appears not specifically designed for the characteristics of infrared images.

(2) The distinctions and advantages of the proposed method compared to existing approaches are not clearly demonstrated.

(3) The iterative degradation removal approach seems functionally similar to cascading multiple degradation-specific networks. Compared to “all-in-one” methods, this design appears cumbersome and complex.

(4) This framework cannot address unknown degradations and is limited to enhancing infrared images with predefined degradation types.

---

> ### Author Rebuttal · Authors · 2025-07-31
>
> ## **Rebuttal to Reviewer RuCv**
>
> We sincerely thank the reviewer for the positive feedback and for recognizing the novelty of our method, the benchmark construction, and the clarity of our presentation. We address the concerns in detail below.
>
> ### 1. **Lack of Infrared-Specific Design (W1, Q1):**
>
> **We argue that our framework is specifically designed to address the complex degradations found in thermal infrared (TIR) images by adopting general prompt-based techniques.** These degradations, including varieties of fix pattern noises and low contrast, **are fundamentally different from those in RGB images and have been insufficiently explored in existing literature.** Prior methods often assume simple or isolated degradations, while our method **introduces a progressive prompt fusion network that enables progressive removal in composite degradation patterns or directly handle single degradation scenario.** In addition, we establish **a dedicated TIR benchmark (HM-TIR) and design a selective progressive training strategy to improve robustness across varying degradation scenarios.** While our approach can apply more broad domain in principle, it provides a targeted solution to a domain-specific problem that is not well addressed by existing models.
>
> ### 2. **Insufficient Comparison with Existing Methods (W2):**
>
> The key different is that, instead of modeling all degradations jointly within a single network, focusing on specific degradation, or using existing all-in-one approaches that address multiple single degradations, **our method employs a prompt-guided framework with a unified backbone and fusion of degradation and scenario-specific prompts.** This design enables explicit and progressively handling of diverse degradations across different scenarios. For comparison, our manuscript includes extensive evaluations demonstrating the effectiveness of our method. For example, the Figure 4 shows our clear advantage over TIR-specific and RGB all-in-one models on synthetic composite data. Figure 5 and Table 2 extend this to real-world TIR images, where our method consistently achieves better visual and metric results. Figure 7 and Table 3 compare baselines with and without our prompt fusion module. These comparisons demonstrate the novelty of our model over TIR-specific methods, RGB all-in-one models, and baseline variants.
>
> ### 3. **Architectural Complexity and Redundancy (W3, Q2):**
>
> Although our method performs iterative processing, it does not cascade multiple independent networks. **Instead, it employs a unified backbone across iterations, modulated by degradation and scenario prompts, enabling progressive removal of each degradation type in different scenarios.** This design avoids structural redundancy, and all iterations reuse the same network parameters, with only lightweight prompt modules introduced. While iterative inference may require more steps than single-pass baselines, the added cost is slight and results in significantly improved performance, as demonstrated in Figure 7, Table 3, and Table 5.
>
> ### 4. **Limited Generalization to Unknown Degradations (W4, Q4):**
>
> We acknowledge that our current framework is based on a predefined set of degradation types commonly observed in TIR images, which may limit its ability to handle unknown degradations. **However, it is important to note that we have already evaluated our method on real-world TIR data (Figure 5, Table 2), demonstrating its strong practical effectiveness.** Additional experiments on other real-world degradations are provided in the supplementary material, primarily to demonstrate that our strategy can effectively enhance TIR images in broader real-world scenarios. **The results further support the effectiveness and generalization capability of our method, with consistent visual improvements in noise reduction, contrast enhancement, and structure preservation.** Given the already significant visual improvements and performance gains, we believe the **current results sufficiently validate the robustness and practicality of our approach.** While our current setup enables effective TIR enhancement, we agree that improving generalization to unknown degradations or adapting to newly emerging degradation types remains a critical and promising direction for future work.
>
> ### 5. **Ambiguity in Prompt Design (Q3):**
>
> We acknowledge that the current manuscript lacks a clear explanation of how the prompts are defined, which may have caused confusion. In our framework, **two types of prompts are used as conditional inputs to guide the network. These prompts are randomly initialized and fixed for each prompt type, and during training,** they gradually encode task-relevant conditions through model optimization under specific prompts. **We manually define the prompt corresponding specific degradation conditions.** Specifically, the prompts are designed to represent the degradation type and processing scenario. As part of future work, we plan to explore more flexible and scalable prompt designs. In particular, we will investigate the use of image-based self-prompting mechanisms, where the model dynamically generates degradation-aware prompts from the input itself. Additionally, we are interested in integrating language-based prompts to express complex degradation descriptions in a more interpretable and user-controllable manner.

---

> > ### Comment · Reviewer_RuCv · 2025-08-04
> >
> > I appreciate the author’s rebuttal, which has satisfactorily resolved my concerns. Accordingly, I have decided to increase my score.

---

### Official Review · Reviewer_3U2T · 2025-07-01

**Clarity:** 2
**Significance:** 3
**Originality:** 3
**Rating:** 4
**Confidence:** 4

**Summary:**

The paper proposes the Progressive Prompt Fusion Network (PPFN) for enhancing thermal infrared (TIR) images, which utilizes type and degradation-specific prompts along with a Selective Progressive Training (SPT) mechanism to adaptively guide the model in handling both single and composite degradations. Extensive experiments demonstrate that PPFN significantly outperforms existing methods.

**Questions:**

(1)	SPT assumes that the degradation process occurs in a specific order. If the actual degradation sequence does not match the assumed order, the model’s performance may be affected.

(2)	If the infrared image simultaneously contains stripe noise, optical noise, and Gaussian noise, how does SPT handle these three types of degradation?

(3)	The paper does not specify how the two prompts are obtained.

(4)	What are the imaging differences between TIR and RGB images? Why would methods designed for the RGB image domain introduce artifacts, and how does the proposed method address these differences?

**Ethical Concerns:**

["NO or VERY MINOR ethics concerns only"]

**Final Justification:**

Thank you to the author for your detailed reply, which effectively answered my questions. After considering the opinions of other reviewers, I have decided to increase my score.

**Limitations:**

yes

**Quality:**

2

**Strengths And Weaknesses:**

Strengths:

(1)	The PPFN establishes prompt pairs based on the thermal imaging process to better address specific degradations under single or multiple conditions.

(2)	A Selective Progressive Training (SPT) mechanism is proposed to refine the model's ability to process complex degradations iteratively.

(3)	The paper also establishes a high-quality, multi-scenario TIR benchmark (HM-TIR) to facilitate comprehensive evaluation.

Weaknesses:

(1)	SPT assumes that the degradation process occurs in a specific order. If the actual degradation sequence does not match the assumed order, the model’s performance may be affected.

(2)	If the infrared image simultaneously contains stripe noise, optical noise, and Gaussian noise, how does SPT handle these three types of degradation?

(3)	The paper does not specify how the two prompts are obtained.

(4)	What are the imaging differences between TIR and RGB images? Why would methods designed for the RGB image domain introduce artifacts, and how does the proposed method address these differences?

---

> ### Author Rebuttal · Authors · 2025-07-31
>
> ## **Rebuttal to Reviewer 3U2T**
>
> We thank the reviewer for their positive feedback and for recognizing the effectiveness of our proposed framework in addressing complex degradations in TIR images. Below, we provide detailed responses to the concerns regarding degradation order, noise handling, prompt design, and TIR-RGB differences.
>
> ### 1. **Fixed Degradation Order Assumption (W1):**
>
> **We argue that in TIR imaging, degradations typically follow a consistent order driven by imaging processes.** Low contrast comes first from limited thermal radiation, followed by blur from defocus or turbulence, and finally hybrid noise such as fixed pattern noise and random noise. Our framework follows this order, and we evaluate on real-world TIR images, **as shown in Figure 5 and Table 2, this design leads to strong performance in real-world scenes.** To test robustness, we also evaluated the model under incorrect degradation removal orders. As shown in Figure 10, performance drops significantly in some case, confirming the model works with a fixed processing order. In future work, we plan to model and predict degradation order to handle more complex scenarios.
>
> ### 2. **Handling Simultaneous Multiple Noise Types in Infrared Images (W2) :**
>
> Regarding diverse noise patterns, we observe that stripe noise and optical distortion often coexist in real-world thermal infrared images due to the radiation inside the imaging body and environmental temperature.[1] **Consequently, rather than handling each type of noise independently, we consider their coexistence as a unified degradation pattern and model it as hybrid noise (see Figure 2(a), Noise Part).** A degradation prompt is employed to guide the model in conditional learning to suppress this pattern during both training and inference. The specific degradation simulation formulations are provided in Supplementary Material A1.1.1.
>
> ### 3. **Ambiguity in Prompt Design (W3) :**
>
> We acknowledge that the current manuscript lacks a clear explanation of how the prompts are defined, which may have caused confusion. In our framework, **two types of prompts are used as conditional inputs to guide the network. These prompts are randomly initialized and fixed for each prompt type, and during training,** they gradually encode task-relevant conditions through model optimization under specific prompts. **We manually define the prompt corresponding specific degradation conditions.** Specifically, the prompts are designed to represent the degradation type and processing scenario. As part of future work, we plan to explore more flexible and scalable prompt designs. In particular, we will investigate the use of image-based self-prompting mechanisms, where the model dynamically generates degradation-aware prompts from the input itself. Additionally, we are interested in integrating language-based prompts to express complex degradation descriptions in a more interpretable and user-controllable manner.
>
> ### 4. **Differences Between TIR and RGB Imaging and Artifact Causes (W4):**
>
> **Unlike RGB imaging, which relies on reflected light and is sensitive to illumination and weather, TIR imaging captures emitted thermal radiation and remains stable under varying conditions.** However, due to its longer wavelength and sensor characteristics, **TIR images suffer from unique degradations such as stripe noise, optical noise, and radiation-caused low contrast, especially in uncooled CMOS-based systems.[1-2]** These structured and composite degradations are uncommon in RGB images and cannot be effectively handled by RGB-oriented models, **which typically focus on random noise, bad weather,  or low illumination and often fail under TIR scenarios, as shown in Figure 4 and 5.** Moreover, modeling all degradations within a single network still yields unsatisfactory results when dealing with complex, dynamic, and compound degradations, as shown in Figure 7 and Table 3.  Consequently, we propose a TIR-specific framework that explicitly and decomposed models thermal degradations using prompt-based learning. Our method integrates a prompt fusion network and sequential tuning strategy to effectively handle both single and composite degradations, demonstrating robust performance where RGB-based models fall short, as demonstrated in Figure 4, Figure 5, and Table 2.
>
> ---
>
> [1] Cai et al. Exploring video denoising in thermal infrared imaging: physics-inspired noise generator, dataset and model. IEEE TIP (2024).
>
> [2] Danaci and Akagunduz. A survey on infrared image & video sets. Multimedia Tools and Applications (2024).

---

> > ### Author Response · Authors · 2025-08-06
> >
> > Dear Reviewer 3U2T,
> >
> > We sincerely appreciate your valuable feedback and hope our responses have addressed your concerns. As the discussion period nears its end, please let us know if you have any remaining questions or suggestions. We are happy to clarify and continue the dialogue.
> >
> > Best regards,
> >
> > Authors of Submission 20148

---

> > > ### Comment · Reviewer_3U2T · 2025-08-08
> > >
> > > Thank you to the author for your detailed reply, which effectively answered my questions. After considering the opinions of other reviewers, I have decided to increase my score.

---

### Note · Authors · 2025-08-16

Dear reviewers and AC,

We sincerely appreciate all the reviewers for their valuable time and insightful comments on our submission. We greatly appreciate the positive feedback on our proposed PPFN module, the SPT training mechanism, and our high-quality infrared dataset.

We have carefully addressed all suggestions, including clarifying the main imaging differences between RGB and TIR images, adding more technical details on the degradation order assumption and prompt design, expanding the related works on degradation simulation pipelines and "all-in-one" image restoration models, and providing a more thorough limitations analysis regarding the prompt formulation, degradation handling capability, and simulation under extreme conditions. All reviewers acknowledged our responses and expressed a consistently positive attitude toward our revisions. We believe these revisions have significantly improved the quality and clarity of our paper, and we thank the reviewers again for their valuable efforts!

Best regards,

Authors of Submission 20148

---

### Decision · Program_Chairs · 2025-09-17

**Decision:**

Accept (poster)

**Comment:**

The paper introduces the Progressive Prompt Fusion Network (PPFN) and a Selective Progressive Training (SPT) mechanism to enhance thermal infrared (TIR) images under complex and composite degradation scenarios.

Key innovations include:

•	Dual prompts (type-specific and degradation-specific) to identify and decouple degradations.

•	A plug-and-play fusion block to integrate prompts into the learning process.

•	A progressive training strategy to iteratively address multiple degradation types.

•	Construction of a high-quality, multi-scene TIR dataset for benchmarking.

Strengths highlighted by reviewers:

•	Novelty and effectiveness in handling hybrid degradations.

•	Superior performance over state-of-the-art methods in both quantitative and qualitative evaluations.

•	Robustness and generalization in real-world scenarios.

•	Clear and satisfactory rebuttal that addressed reviewers' concerns, leading to score increases from multiple reviewers.